# The Willingness to Pay in the Food Sector. Testing the Hypothesis of Consumer Preferences for Some Made in Italy Products

**Lucio Cappelli [1],\*, Fabrizio D'Ascenzo [2], Maria Felice Arezzo [2], Roberto Ruggieri [2] and Irina Gorelova [2]**

[1]  Department of Economics and Law, University of Cassino and Southern Lazio, 03043 Cassino, Italy
[2]  Faculty of Economics, University of Rome "La Sapienza", 00161 Rome, Italy;
    fabrizio.dascenzo@uniroma1.it (F.D.); mariafelice.arezzo@uniroma1.it (M.F.A.);
    roberto.ruggieri@uniroma1.it (R.R.); irina.gorelova@uniroma1.it (I.G.)
\*  Correspondence: cappelli@unicas.it; Tel.: +39-335-331126

**Abstract:** Previous publications have shown that Italian consumers are willing to pay a premium price for certain categories of Made in Italy products. The premium price has proven to be higher in the food sector. This study provides an extensive literature review on the topic and aims to test a hypothesis regarding consumer preferences towards some Made in Italy food products of mass consumption (olive oil, meat and fish), with specific reference to the value systems that influence the purchase. This paper studies the correlation between the potential willingness to pay a premium price for the mentioned products and the characteristics of consumers' sample. The results obtained confirm the willingness to pay for Made in Italy products and correlate the willingness to pay a premium price with the level of education of the respondents to the questionnaire. Thus, these findings show that consumers with a higher educational level tend to make more sustainable food choices and by doing so lean toward a sustainable lifestyle.

**Keywords:** Made in Italy; willingness to pay; country of origin; quality; sustainable food choices

## 1. Introduction

This paper contributes to an extensive study on Made in Italy and to both sustainable business strategies and public policies that are aimed at the promotion of "Made in" initiatives. The overall analysis is based on the following two research questions: (1) is there a preference in terms of consumption for Made in Italy products over similar non-Made in Italy products? And if this preference is demonstrated, (2) is there a willingness to pay, in terms of quantity, a premium price for such Made in Italy products?

This research is trying to reveal whether the preference for Made in Italy products (which for some product segments appears to be almost obvious and well supported by the literature) also turns into an effective willingness to pay premium prices.

The main results obtained so far by Cappelli et al. (2016, 2017, 2019) [1–3] have clearly shown that Made in Italy products not only represent a well-defined conceptual category in the minds of Italian consumers, but that there is a premium price for Made in Italy products in the studied sectors (food, furniture, textile/clothing, mechanical automation), within a range of 10–30% depending on a sector. The results obtained so far [3] have shown that the premium price is generally higher for Made in Italy products in the food sector. For this reason, this particular study is aimed at deepening the empirical investigation focusing on some specific consumer food products: extra virgin olive oil, pre-packaged meat and raw fish, including seafood. The goal of this specific research is not

one of quantifying the willingness to pay a premium price, but to study the correlation between the characteristics of consumers' sample and the willingness to pay a premium price (at least 10%) for the mentioned products.

This paper proposes a review of the literature, focusing on the scientific strand concerning the 'willingness to pay', specifically for Made in Italy olive oil, meat and fish products. Then the original results from the empirical investigation are presented.

## 2. Literature Review

There is a lack of sufficient studies in the literature for a full understanding of the real value of Made in Italy in terms of Italian consumers' willingness to pay. In fact, there are still few studies that quantify the willingness to pay of consumers towards Made in Italy products. This is true both in general and specifically in the food sector for products considered in this paper: extra virgin olive oil, meat and fish.

By summarizing the results of the literature research on the matter under inquiry, a narrative literature review was conducted, considering the contributions from the period from 2008 to 2020. The authors found it appropriate to consider the initial year of the international economic and financial crisis as a starting date of the literature review because it had a direct and negative impact on the consumption of alimentary products by Italians [4]. The analysis of 49 selected items (13 for meat, 19 for extra virgin olive oil and 17 for fish) reveals the relevance in consumer preferences, especially in the following product attributes: certification of geographic origin, food safety, sensory characteristics, brand and price.

Almost all the authors highlight a correlation of the aforementioned attributes with the willingness to pay a premium price, but only a few propose a quantitative estimate in their empirical investigations. Studies of Made in Italy olive oil have revealed that the individual consumer choice for paying a premium price is particularly linked to elements such as the area of origin and the information on the label, specifically the quantity of polyphenol concentration [5]. Sustainable production processes are as important as the country of origin for the consumers [6]. Consumers are willing to pay a premium price of around 50% if the protected geographical indication and the protected designation of origin are stated on the label [7]. The willingness to pay increases when the Italian origin of the product is specified [8]. Several studies show [9–11] that consumers are more likely to purchase extra virgin olive oil with a premium price basing their decision on the organoleptic characteristics and environmentally friendly production processes in addition to the country of origin. Three groups of scholars [12–14] state that consumers pay a premium price if the olives are Made in Italy, i.e., if the origin is 100% Italian and if the production is local. From the several studies [15,16] the regional origin of the oil emerges as the fundamental characteristic in order to have a premium price. Through extensive empirical research it was proved that the premium price for Made in Italy oil is €3.76/L more if it is organic [17]. Other scholars [18] estimate that the price customers are willing to pay for a Made in Italy oil is €4.00/L and that the decision to pay a premium price is also motivated by the presence of indications concerning in particular the level of acidity of the oil and the general health claim. Italian consumers are willing to pay €6.1/L for Made in Italy olive oil, and organic label is another important value for the customers [19]. The study of the Institute of Services for the Agricultural Food Market (ISMEA 2018) [20] shows that the interest and conditions for the purchase of Made in Italy oil vary between Italian geographic areas. The aforementioned report shows a preference for the purchase of oil with Italian origin; consumers from Northern Italy are willing to spend up to €20 more for a bottle of oil with the territorial indication displayed and prefer oils from Tuscany, Liguria and Puglia. Consumers from the central part of Italy are willing to pay up to three times the basic price of a bottle of extra virgin olive oil if the Italian origin is guaranteed and if there is information on the region of origin on the label. In Southern Italy, consumers prefer to buy oil from friends or acquaintances and are willing to pay a double price for oil of Italian origin. Research conducted among consumers from Veneto [21] has shown that they pay more attention to the place of the oil production. This attribute, however, was

found to be less important in areas where olive cultivation occupies only a small part of the cultivated area. In this case, consumers tend to choose oil from other regions where production is widespread. Most of the research respondents are willing to pay a premium price for the purchase of oil of Italian or Venetian origin and if it is labeled as such. A study on the preferences of oil consumers in the Abruzzo region showed that the oil production region is a very important attribute for consumers [22]. The oil of Abruzzo origin was ranked first among the purchasing preferences and since "local production" was the most relevant quality attribute, it is likely that, for the majority of respondents, regional patriotism will play an important role in their choice of oil. Analyzing the preferences for the olive oil from the neighboring regions (North, Central and South Italy), research has shown that the areas geographically close to Abruzzo (Central Italy) obtained a higher preference score from the interviewees than more distant areas. The expected average willingness to pay a premium price for Abruzzo oil (1 L) was €9.45 (±€2.67), Tuscany: €8.74 (±€2.69) and Puglia: €7.94 (±€3.42). The origin of the oil has a strong impact on the price, with over 50% of premium prices for oil from northern regions and 24% premium prices from central regions, compared to those of Southern Italy [23]. However, the presence of a certification of origin has not significantly influenced the willingness to pay a premium price.

The review of the literature concerning the willingness to pay for Made in Italy meat shows that there are fewer studies related to the quantitative determination for the willingness to pay a premium price. In general, studies have shown that the most important attribute associated with the premium price concerns the indication "animal welfare", which means respecting methods of cattle, pig and sheep breeding. The premium price ranges from 10% to 20% for Made in Italy meat if the label expressly states "animal friendly" or "animal welfare" [24,25]. The studies of several scholars [26–28] at a qualitative level confirm the importance of the sustainability of the production processes and the territorial origin of the meat. Other studies have shown an opposite result. The local origin of beef is less important than price, "animal welfare", animal breed, brand and organic label for the consumers from the northwest of Italy [29]. Italian consumers who pay attention to the sensory aspects of meat products are not interested in the Italian origin of the product [30]. The label seems to be a valid and important source of information, nutritional and non-nutritional, which determines the willingness to pay a premium price for these products [27,31–36].

The review of the literature concerning the Made in Italy fish sector reveals some papers that present quantitative and qualitative outcomes on the willingness to pay a premium price. Several international studies have shown that consumers prefer local fish to imported fish and are willing to pay an adequate premium price for local products [37–39]. However, the willingness to pay a premium price for Made in Italy products by Italian consumers is also influenced by environmental sustainability factors. The eco-labels of seafood could increase the willingness to pay of Italian consumers by between 16% and 24%, highlighting how premium prices can become an effective tool for sustainable resource management [40]. However, an inverse trend also exists: although Italians consider environmental protection and the conservation of marine habitats an important issue, they are not willing to pay a higher price to support sustainable and selective fishing [41]. Domestic and local provenance of fish is particularly important for Italian consumers and a willingness to pay around 14% more on average for fish produced locally (within the region) is demonstrated [42]. The extensive literature review [43] shows that consumers have a positive perception of sustainable fish products and willingness to pay the premium price. On a qualitative level, there is an interesting study [44] that presents a systematic review of the literature examining around 50 national and international articles in order to evaluate the purchasing behavior of consumers towards fish products. The studies indicated above show the relevance in consumer preferences of the following product attributes: certification of geographical origin, label and sustainable production. The estimated premium price is +€3.9/kg [45] and +€0.86/kg [46]. For some authors, the willingness to pay increases if the consumer finds all the clearly defined information on the label [47], in particular if the premium price has been estimated at +€1.24/1.25 per kg [46,48]. If the Made in Italy fish label is organic, the premium price is +€2.76/kg [49]. Furthermore, the technical report of a project commissioned by the European Union [50] describes

that the willingness to pay a premium price by Italian consumers for the fish product labeled Made in Italy is between 10% and 15% more if the product meets requirements regarding sustainability, organicity and production with respect for animal welfare. The better quality of the products increases the willingness to pay a premium price [51]. Other authors [52,53] in their research highlight how the direct relationship between safety and sustainability affects consumer choices and add the additional attributes such as the country of origin of the fish product and the presence of the eco-label.

## 3. Materials and Methods

The consumers' preferences were measured by means of a questionnaire administered to a random sample of 410 respondents (female (53%), male (47%), aged 18 to 66 years) interviewed during October 2019 while exiting supermarkets of a well-known food distribution chain in Rome.

A binary logistic regression modelling approach was used to empirically estimate the willingness to pay a 10% markup for Made in Italy products. Since this paper did not set the goal of quantifying the willingness to pay a premium price, but to test the hypothesis that some characteristics of the sample may influence the willingness to pay, the decision to apply a 10% markup was based on previous studies. As the extensive literature review has shown, all the studied groups of products have different quantitative value of the willingness to pay. Previous research has shown that the consumers are willing to pay a premium price of around 50% for Italian olive oil [7], the premium price for Made in Italy meat ranges from 10% to 20% [24,25] and for fish from 10–20% [50] to 24% [40]. In this regard, a 10% markup is the lowest premium price value for all the groups of products.

The dependent variable takes value 1 if the respondent is willing to pay a price at least 10% higher for Made in Italy products and 0 otherwise.

The independent variables consist of a set of socio-demographic information of the respondent (gender, age, educational qualification, profession), a binary variable which takes value 1 if the respondent lives in the neighborhood and 0 otherwise, and a set of variables that can frame the consumers beliefs: the attitude towards consumption and Made in Italy (on a Likert scale from very much agreeing to not at all agreeing); the willingness to pay premium prices for the Made in Italy product; knowledge regarding the quality of the production chain of the products purchased; the main source of knowledge/information. Most of the questionnaire items were frequently mentioned in the scientific literature and have already been tested [54–57]. Table 1 illustrates both the dependent variable and the covariates.

**Table 1.** Variable list for empirical modelling.

| Variable | Variable Type | Values |
|---|---|---|
| *Dependent variable:* | | |
| W: Willingness to pay | Binary | 1 = Yes, I am willing to pay a premium price<br>0 = No, I am not |
| *Independent variables:* | | |
| Gender | Binary | 1 = female |
| Age | Discrete grouped in classes | 18–24<br>25–30<br>31–40<br>41–50<br>51–60<br>61 or more |
| Education | Categorical | 1 = secondary school or lower<br>2 = university degree or higher |
| Profession | Categorical | 1 = student/unemployed<br>2 = housewife<br>3 = workman/employee<br>4 = employer<br>5 = retired |

**Table 1.** *Cont.*

| Variable | Variable Type | Values |
|---|---|---|
| Neighborhood | Binary | 1 = Yes, I live close to the supermarket.<br>0 = No, I live in a different neighborhood. |
| D1: Before I buy a product, I compare prices and I buy the cheapest.<br>D2: I only buy products of a specific brand I trust.<br>D3: I buy those products I have had a positive consumer experience with.<br>D4: I base my purchases on the information reported in the product tags.<br>D5: I buy the products based on their origin.<br>D6: I am more inclined to purchase Made in Italy.<br>D7: I do prefer Made in Italy even if it is more expensive.<br>D8: I buy Made in Italy because it is an ingrained habit.<br>D9: I buy Made in Italy because of the product quality.<br>D10: I think it is justified that Made in Italy is more expensive. | Likert scale | 1 = fully disagree<br>2 = quite disagree<br>3 = quite agree<br>4 = fully agree |

Since our goal is to predict the probability that an individual is willing to pay a premium price to buy a Made in Italy product and to understand which socio-demographic characteristics increase this probability we decided to make use of a logistic model. Therefore letting $W_i$, $(i = 1, 2, \ldots, n)$ be a binary outcome (0/1) for the *i*th subject, which follows a Bernoulli distribution with the probability $\pi_i = \Pr(W_i = 1)$, the logistic regression model can be defined as

$$logit(\pi_i|X_i) = \left( \frac{\pi_i}{1 - \pi_i}|X_i \right) = \eta_i = X_i \varphi \tag{1}$$

where $\varphi^T = \left[ \beta_0, \beta_1, \ldots, \beta_q \right]$ is the vector of regression coefficients of length $(q + 1)$, and $X_i$ is the *i*th row vector of the predictor matrix $X$ which has the order $n \times (q + 1)$. The term $\eta_i$ is called the risk score.

Furthermore, since the interviewed customers are likely to live close to the supermarket where they had shopped and since the behavior of consumers residing in the same neighborhood (cluster) is expected to be alike, the estimates of the standard error need to take into account the dependencies within clusters.

In this scenario, we used the clustered version of a Huber–White sandwich estimator of the variance [58]. Since clusters are independent from each other, we can sum the scores within a cluster to create a "super observation" and then use the standard formula for the robust sandwich estimator on these independent super observations. The robust variance estimator is:

$$\text{Var}(\hat{\beta}) = D \left[ \frac{n_C}{n_C - 1} \sum_{i=1}^{n_C} \left( \sum_{j \in C_i} u_j \right)^T \left( \sum_{j \in C_i} u_j \right) \right] D \tag{2}$$

where $D = -H^{-1}$ is the traditional covariance estimate (i.e., minus the inverse of the Hessian matrix), $u_j = \frac{\partial \ln L(\beta, y_j x_j)}{\partial \beta}|_{\hat{\beta}}$ is the derivative of the log likelihood for the *j*th observation evaluated at $\hat{\beta}$ and $n_c$ is the number of observations in cluster C [59] (see [58–60] for further details). All the estimates were done using Stata 13 [60] and are reported in Table 2 in the following section.

**Table 2.** The estimates of the logistic regression model coefficients.

| Logistic Regression<br>Number of Observations: 410<br>Log Pseudolikelihood = −209.388<br>Pseudo $R^2$ = 0.283<br>(Std. Err. Adjusted for 5 Clusters in Neighborhood) | | |
|---|---|---|
| | **Coef.** | **Pr > \|z\|** |
| *D1: Before I buy a product, I compare prices and I buy the cheapest (ref: fully agree).* | | |
| quite agree | 0.625 | 0.003 |
| quite disagree | 1.52 | 0.000 |
| fully disagree | 0.92 | ns |
| *D2: I only buy products of a specific brand I trust (ref: fully agree).* | | |
| quite agree | −0.021 | ns |
| quite disagree | −0.188 | ns |
| fully disagree | −0.195 | ns |
| *D3: I buy those products I have had a positive consumer experience with (ref: fully agree).* | | |
| quite agree | −0.117 | ns |
| quite disagree | −0.399 | ns |
| fully disagree | 2.73 | 0.018 |
| *D4: I base my purchases on the information reported in the product tags (ref: fully agree).* | | |
| quite agree | 0.392 | ns |
| quite disagree | 0.251 | ns |
| fully disagree | −0.477 | ns |
| *D5: I buy the products based on their origin (ref: fully agree).* | | |
| quite agree | 0.21 | ns |
| quite disagree | −0.615 | ns |
| fully disagree | −0.254 | ns |
| *D6: I am more inclined to purchase Made in Italy (ref: fully agree.)* | | |
| quite agree | 0.169 | ns |
| quite disagree | 0.599 | ns |
| fully disagree | 0.412 | ns |
| *D7: I do prefer Made in Italy even if it is more expensive (ref: fully agree).* | | |
| quite agree | −0.973 | 0.000 |
| quite disagree | −1.217 | 0.013 |
| fully disagree | −1.31 | |
| *D8: I buy Made in Italy because it is an ingrained habit (ref: fully agree).* | | |
| quite agree | −0.016 | ns |
| quite disagree | −0.431 | ns |
| fully disagree | −0.649 | ns |
| *D9: I buy Made in Italy because of the product quality (ref: fully agree).* | | |
| quite agree | −0.693 | 0.000 |
| quite disagree | −0.568 | ns |
| fully disagree | 0.165 | ns |
| *D10: I think it is justified that Made in Italy is more expensive (ref: fully agree).* | | |
| quite agree | −0.353 | ns |
| quite disagree | −1.254 | 0.000 |
| fully disagree | −1.847 | 0.000 |
| *Gender: female* | −0.445 | ns |
| *Age (ref: 18–24)* | | |
| 25–30 | −1.117 | 0.023 |
| 31–40 | −0.897 | ns |
| 41–50 | −0.808 | ns |
| 51–60 | −1.531 | ns |
| 61 or more | −1.623 | ns |
| *Education (ref: graduated)* | | |
| University degree or higher | 0.515 | 0.000 |
| *Lives in the neighborhood* | 0.469 | ns |
| *Profession (ref: student/unemployed)* | | |
| housewife | 0.543 | ns |
| workman/employee | 0.737 | ns |
| employer | 0.189 | ns |
| retired | −0.115 | ns |
| cost | 1.132 | ns |

ns: not significant at 5% level.

## 4. Results

The model tests the hypothesis of the willingness to pay an additional 10% based on the responses to the questionnaire. If $P$ ($p$-value) is greater than 0.05 the variable is not significant. The positive value of the coefficient increases the probability of the event occurring (i.e., the purchase of the Made in Italy products with a willingness to pay a premium price of more than 10%).

The results of the survey are presented in Table 2.

The output of the survey did not show significant results for the whole set of research questions, but for many of them there are interesting findings for further discussion.

## 5. Discussion

According to the model, focusing only on the variables in the table that have a relevant impact on the willingness to pay, it is possible to propose the following main considerations.

Q1. The purchase decision with a willingness to pay more than an additional 10% does not seem to be preceded by a price comparison activity. This suggests that Made in Italy buyers with the willingness to pay are less sensitive to the price factor when they purchase the products.

Q3. Positive consumer experience does not seem essential to increasing the probability of purchasing the Made in Italy product with the willingness to pay more than an additional 10%. This result suggests that the probability of purchasing Made in Italy products with a willingness to pay a premium price is not linked to previous consumer experience but to other factors.

Q5. The territory of origin appears to play an important role in increasing the probability of purchasing Made in Italy products with a willingness to pay a premium price. This is consistent with the expectation that the place of origin of the product will impact consumers' purchasing decisions.

Q7. The preference for Made in Italy products is definitely related to the probability of purchase with willingness to pay a premium price. This consideration is perfectly in line with expectations of this research.

Q10. The probability of purchasing a Made in Italy product with a willingness to pay more increases based on the fact that a higher price is considered justified for a Made in Italy product compared to a non-Made in Italy product. This result is particularly interesting as it seems to suggest that a higher price for Made in Italy products is not only "assumed as such" and "borne" by consumers, but is considered to be "legitimate" and "correct".

Regarding the low significance of the age variable, the younger age group (18–24) shows a lower probability of the willingness to pay a premium price for Made in Italy products. This result may have two interpretations. On the one hand, these data can be interpreted in terms of a lower financial freedom of the considered age group and hence their lower spending capacity. On the other hand, the analysis of the answers to the subsequent questions, which showed a strong relationship between higher educational qualifications and probability of the willingness to pay a premium price for Made in Italy products, makes the age factor a negligible variable.

The possession of a university degree or a post-graduate degree increases the probability of purchasing Made in Italy products with the willingness to pay a premium price. These results correspond with the findings in previous studies of different Made in Italy products [23,30,61,62]. The type of profession does not seem to influence the willingness to pay for Made in Italy products. The analysis of the answers to the two previous questions is probably the most interesting aspect that emerges from the results of the research. In fact, it is evident that the variable of education strongly influences the probability of purchasing Made in Italy products with a willingness to pay a premium price, while the profession and the consequent greater or lesser economic well-being do not show the same effect.

## 6. Conclusions

The present study has its limitations in the random audience sampling. Another limitation is the fact that the choice of examined products did not fully correspond to the ones studied in the literature review. Still, the conducted study helps to reveal the main features of the consumers who are ready to pay a premium price for Made in Italy food products. One of the hypothesis of the research was that the level of education influences consumer preferences for Made in Italy products. Indeed, this research has revealed that the educational factor represents the element that most of all characterizes the typical consumer with the willingness to pay a premium price of over 10% more for purchasing Made in Italy products and in doing so making a more sustainable choice. This is a particularly interesting finding on both the scientific and managerial levels. If consumers' willingness to pay is based upon their education, it is expected that a greater level of education regarding the importance of sustainability will lead to a greater willingness to pay for better sustainability practices. This is extremely interesting to study in future research.

**Author Contributions:** Conceptualization, L.C., F.D., M.F.A., R.R. and I.G.; methodology, L.C., R.R., M.F.A. and I.G.; software M.F.A. and I.G.; formal analysis, M.F.A.; investigation, L.C., R.R. and I.G.; data curation, M.F.A.; writing—original draft preparation, L.C., R.R., M.F.A. and I.G.; writing—review and editing, L.C., R.R. and I.G.; supervision, F.D., L.C. and R.R. All authors have read and agreed to the published version of the manuscript.

**Funding:** This research received no external funding.

**Conflicts of Interest:** The authors declare no conflict of interest.

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
