# Peer review of "The Willingness to Pay in the Food Sector. Testing the Hypothesis of Consumer Preferences for Some Made in Italy Products"

_sustainability, doi:10.3390/su12156275_

Round 1

Reviewer 1 Report

Reading the title, I was expected to find out the respondent's mean WTP or consumer’s willingness to pay an extra amount for some food products. However, the paper is simply examining a hypothesis, without estimating an amount. So, I don't think the title is accurate.

Νo details and documentation are used for the sampling technique they use.

Authors said that “The consumer’ preferences were measured by mean of a questionnaire”, but they did not use any econometric model to estimate the mean WTP.

Indeed for predicting dichotomous outcomes, logistic regression has been revealed as the appropriate statistical technique. However, I am not sure about the use of Equation (1) for the logistic regression model. Please provide more documentation.

I consider that only the section “literature review” has been sufficiently developed. In the rest, there is a substantial deficit. “Materials and Methods” section need to be improved. The paper lacks a Discussion Section which is a major shortcoming.  This is reflected in the extent of each section, as well.

Last, how this study is linked to journal area (Sustainability) shall be better communicated.

I regard that references are not written properly. In the text, only the numbers should be placed in square brackets.

Some punctuation marks are missing (for example in the abstract).

Reviewer 2 Report

The paper “The Willingness to Pay for Made in Italy. Empirical Research on Some Products in the Food Sector” deals with an interesting topic using a not so innovative technique. The overall quality of the work is below average, and an improvement Is needed. Also, I strongly suggest an English revision.

  • The title is not appealing, in my opinion and I suggest authors to rewrite it.
  • The abstract could be improved (“confirming the willingness to pay for Made in Italy products”?)
  • In the whole introduction and literature review, journal’s guidelines have not been followed with attention, authors should write only numbers without citing the paper in the text.
  • The introduction section should be improved. What kind of “meat” do you refer to? What kind of “fish” do you refer to? Be more specific please.
  • Line 35-36. Which studies are you referring to?
  • Literature review is well written and presented and all the relevant papers have been cited.
  • After a very well written literature review, Section 3 is very poor if compared to the previous section. I would suggest authors to improve the description of the experiment and provide some information regarding socio-demographic characteristics of their sample. For example, table 1 could be replaced with a table regarding the description of the sample and the actual table 1 could be moved to appendix.
  • As regards independent variables, authors write “…set of variables that can frame the consumers beliefs: the attitude towards consumption and Made in Italy […]; the willingness to pay premium prices for the Made in Italy product; knowledge regarding the quality of the production chain of the products purchased; the main source of knowledge / information.” Have the items been collected from previous studies or tested before?
  • Please, explain better how the 10% increase has been established.
  • I can see no discussion at all. Please provide at least some references in order to strengthen your conclusion (which is a missing section).
  • Which are the limitations of the study? You should also consider writing some further developments.

Round 2

Reviewer 1 Report

Most of the suggestions have been done.

Author Response

Dear reviewer, We are happy that the changes made to the paper based on your comments have been appreciated.
We proceeded to further reread the paper with a native English language, and made some small changes (47, 141, 249, 265).
Best regards and thanks again.

Reviewer 2 Report

I'm glad that authors addressed almost all my comments. Some minor concerns regards text editing (tables are not so "clean" and understandable). 

Author Response

Dear reviewer, we are happy that the changes made to the paper based on your comments have been appreciated.
We made a small adjustment to table 1, trying to make it more readable.
Best regards and thanks again.